# DISTINGUISHABILITY OF ADVERSARIAL EXAMPLES

## ABSTRACT

Machine learning models including traditional models and neural networks can be easily fooled by adversarial examples which are generated from the natural examples with small perturbations. This poses a critical challenge to machine learning security, and impedes the wide application of machine learning in many important domains such as computer vision and malware detection. Unfortunately, even state-of-the-art defense approaches such as adversarial training and defensive distillation still suffer from major limitations and can be circumvented. From a unique angle, we propose to investigate two important research questions in this paper: Are adversarial examples distinguishable from natural examples? Are adversarial examples generated by different methods distinguishable from each other? These two questions concern the *distinguishability* of adversarial examples. Answering them will potentially lead to a simple yet effective approach, termed as *defensive distinction* in this paper under the formulation of multi-label classification, for protecting against adversarial examples. We design and perform experiments using the MNIST dataset to investigate these two questions, and obtain highly positive results demonstrating the strong distinguishability of adversarial examples. We recommend that this unique defensive distinction approach should be seriously considered to complement other defense approaches.

## 1 INTRODUCTION

Machine learning models including SVMs (Biggio et al., 2013) and especially deep neural networks (Szegedy et al., 2014) can be easily fooled by adversarial examples which are generated from the natural examples with small perturbations. Quite often, both machine learning models and humans can classify the natural examples such as the images of pandas with high accuracy, and humans can still classify the adversarial examples as pandas with high accuracy because the small perturbations are imperceptible; however, machine learning models are fooled to misclassify adversarial examples as some targets such as gibbons (Goodfellow et al., 2015) desired by attackers.

This intriguing property or vulnerability of machine learning models poses a critical challenge to machine learning security, and it impedes the wide application of machine learning in many important domains such as computer vision (e.g., for self driving cars) and even in malware detection (Biggio et al., 2013; Xu et al., 2016). Furthermore, the discovery of new and powerful adversarial example generation methods such as (Szegedy et al., 2014; Goodfellow et al., 2015; Carlini & Wagner, 2017; Chen et al., 2018; Kurakin et al., 2017b; Madry et al., 2018; Dong et al., 2018; Papernot et al., 2016a; Moosavi-Dezfooli et al., 2016; Sabour et al., 2016; Uesato et al., 2018) goes on without cessation, indicating to a certain extent the unlimited capabilities for attackers to continuously and easily fool machine learning models. On the other hand, even state-of-the-art defense approaches such as adversarial training (Szegedy et al., 2014; Goodfellow et al., 2015) and defensive distillation (Papernot et al., 2016b) still suffer from major limitations and can be circumvented (Section 2). Therefore, the unfortunate status quo is that attackers prevail over defenders.

In this paper, from a unique angle, we propose to investigate two important research questions that concern the *distinguishability* of adversarial examples. Question 1: are adversarial examples distinguishable from natural examples? Question 2: are adversarial examples generated by different methods distinguishable from each other? If the answer to Question 1 will be positive, i.e., given a certain classification task such as image classification, generated adversarial examples (regardless of the objects they represent) largely belong to one class while natural examples belong to the other class, then defenders can simply discard those adversarial examples to protect the machine learning

models. If the answer to Question 2 will be positive, i.e., adversarial examples generated by different methods clearly belong to different classes, defenders can better protect the machine learning models, for example, by incorporating the corresponding examples into the adversarial training process to enhance the robustness of the models. Besides such practical benefits, answering these two questions may also help researchers further identify the nature of adversarial examples.

Formally, we consider a classification problem in adversarial environments as a multi-label classification problem. That is, upon seeing a new input such as an image, while the original task such as classifying the image as a certain object is important, it is also important to classify the image as a generated adversarial vs. a natural example in the first place. We formulate this multi-label classification problem in Section 3 to guide us in answering the two questions, and term the corresponding defense approach as *defensive distinction*, which distinguishes adversarial vs. natural examples and distinguishes adversarial examples generated by different methods to protect against the attacks.

We design and perform experiments using the MNIST dataset to investigate the two research questions and evaluate the effectiveness of our defensive distinction approach. In our experiments, we consider multiple scenario-case combinations that defenders either know or do not know the neural network, source images, and methods as well as parameters used by attackers for generating adversarial examples. We obtain highly positive answers to both research questions. For example, in some typical cases, adversarial vs. natural examples can be distinguished perfectly with 100% accuracy, while adversarial examples generated by different methods can be distinguished with over 90% accuracy. Our experimental results demonstrate the strong distinguishability of adversarial examples, and demonstrate the value of the defensive distinction approach. We recommend that this unique defense approach should be seriously considered to complement other defense approaches.

We make four main contributions in this paper: (1) we propose to investigate two important research questions that concern the distinguishability of adversarial examples; (2) we formulate a classification problem in adversarial environments as a multi-label classification problem to answer the two questions; (3) we propose and explore a unique defense approach termed as defensive distinction; (4) we design and perform experiments to empirically demonstrate the strong distinguishability of adversarial examples and the value of our defensive distinction approach.

## 2 RELATED WORK

Adversarial examples can easily fool machine learning models, and they can be generated by a number of powerful methods. One representative method is L-BFGS (Szegedy et al., 2014), which formulates the adversarial example generation problem as a box-constrained optimization problem and uses line-search to identify approximated optimal solutions. Another representative method is FGSM (Fast Gradient Sign Method) (Goodfellow et al., 2015), which efficiently obtains an optimal max-norm constrained perturbation for generating adversarial examples. FGSM inspired researchers to pursue a few fine-grained refinements over it, resulting in powerful methods such as BIM (Basic Iterative Method) by Kurakin et al. (2017b), PGD (Projected Gradient Descent) by Madry et al. (2018), and MIM (Momentum Iterative Method) by Dong et al. (2018). L-BFGS also inspired researchers to design more powerful or general methods such as C&W by Carlini & Wagner (2017) and EAD (Elastic Net Method) by Chen et al. (2018). JSMA (Jacobian-based Saliency Map Approach) (Papernot et al., 2016a) is another representative method, which constructs adversarial saliency maps to guide the selection of input features for perturbation in multiple iterations.

On the defense side, one representative approach is adversarial training (Szegedy et al., 2014; Goodfellow et al., 2015), which basically uses both the generated adversarial examples and the natural (a.k.a clean) examples to train the machine learning models with better regularization. Another representative approach is defensive distillation (Papernot et al., 2016b), which basically uses the same neural network architecture to train a distilled network based on the probability vectors produced by the original network, thus reducing the models' sensitivity to small input perturbations. Both approaches in essence focus on improving the generalization and thus robustness of the machine learning models that perform the original classification task. While representing the state-of-the-art, both approaches still suffer from major limitations and can be circumvented (Goodfellow et al., 2018). For example, defensive distillation was found to be ineffective against C&W attacks (Carlini & Wagner, 2017) and black-box attacks (Papernot et al., 2017), while adversarial training was

found to be ineffective against adversarial examples generated by many iterative methods such as BIM (Kurakin et al., 2017a), PGD Madry et al. (2018), and MIM (Dong et al., 2018).

Overall, powerful adversarial example generation methods exist and will continue to be proposed, while existing defense approaches are far from being perfect yet. The unfortunate status quo is that attackers prevail over defenders. Our work is likely one step toward making a change.

## 3 DEFENSIVE DISTINCTION PROTECTION APPROACH

We now present the problem formulation and the corresponding defensive distinction protection approach; we also define a threat model that can be used for evaluating our approach.

### 3.1 MULTI-LABEL CLASSIFICATION FORMULATION OF THE PROBLEM

To date, a classification problem in adversarial environments is viewed as a single-label classification problem, where a function or model $f : \mathbb{X} \to \mathbb{Y}$ is learned to assign each input example $x_i \in \mathbb{X}$ a single label $y_i \in \mathbb{Y}$, where the range $\mathbb{Y}$ is a set of $m$ singletons representing $m$ possible classes such as 10 possible digits. In other words, a single concept or semantic meaning is associated with each example. State-of-the-art defense approaches such as adversarial training and defensive distillation reviewed in Section 2 do not change this basic problem formulation.

However, considering the two important *distinguishability* questions that we propose to investigate (*Question 1: are adversarial examples distinguishable from natural examples? Question 2: are adversarial examples generated by different methods distinguishable from each other?*), it is straightforward for us to formulate a classification problem in adversarial environments as a multi-label classification problem. That is, a function or model $f' : \mathbb{X} \to \mathbb{Y}'$ is learned to assign each input example $x_i$ a pair of labels represented by $y_i' = (a_i, b_i) \in \mathbb{Y}'$. Here, the new range $\mathbb{Y}' = \mathbb{Y} \times \mathbb{Z}$ is the Cartesian product of $\mathbb{Y}$ (i.e., the range in the original classification task) and $\mathbb{Z}$, where $\mathbb{Z}$ is a set of $n$ singletons representing $n$ possible classes regarding if $x_i$ is an adversarial example. In the simplest situation, $n = 2$ and $b_i$ indicates either an example is natural (i.e., clean) or adversarial. In a more complex situation, $n > 2$ and $b_i$ can indicate either an example is natural or generated by one of $n - 1$ different adversarial example generation methods.

Under this new problem formulation, two concepts or semantic meanings [1] are associated with each example. The solution to the problem will help us answer the two research questions, and help defenders simply discard or leverage the identified adversarial examples to better protect their machine learning models as described in Section 1. Moreover, one major advantage of a multi-label classification formulation and the corresponding multi-label learning is on exploring the intrinsic correlations between multiple labels. In real-world adversarial environments, this advantage is significant because it will allow researchers and defenders to perform in-depth analysis of the behavior of attackers. For example, do attackers create most adversarial examples targeting at some specific classes such as gibbons? do attackers use different generation methods for different targeted classes? and, do attackers use examples of some specific source classes to create adversarial examples targeting at some specific classes (Appendix A)? These and perhaps other types of label correlation analyses will be valuable for inferring the real intentions of attackers, and even for revolutionizing the design of the original machine learning systems.

### 3.2 DEFENSIVE DISTINCTION APPROACH

We simply define the *defensive distinction* protection approach as training a model for a multi-label classification problem formulated in adversarial environments to identify adversarial examples and protect against them. So a training set will include both natural examples and generated adversarial examples with their ground truth $b_i$ values correspondingly labeled in $y_i' = (a_i, b_i) \in \mathbb{Y}'$. The ground truth $a_i$ values for natural examples are obviously known based on some existing dataset such as MNIST. The ground truth $a_i$ values for adversarial examples can be missing (Yu et al., 2014) or not needed (as in our solution described below) depending on specific learning algorithms.

---

[1]More labels could be added to include more concepts or semantic meanings as described in Appendix A.

Many multi-label learning algorithms have been proposed with different strategies or considerations on the order-of-correlations among labels (Zhang & Zhou, 2014). In this paper, we experiment with the simplest first-order strategy that basically transforms a multi-label classification problem into multiple single-label problems. More specifically, three models will be independently trained:

- DDP-Model (defensive distinction primary model): a single-label binary model for distinguishing an input example $x_i$ as either natural or adversarial.

- Original-Model: a single-label multi-class model for the original task on classifying the input example $x_i$ as one of multiple possible objects such as one of 10 digits.

- DDS-Model (defensive distinction secondary model): a single-label multi-class model [2] for distinguishing known adversarial examples as generated by different specific methods.

This simple strategy provides a baseline solution to the defensive distinction approach. It is simple in model formulation, computation, and evaluation, albeit high-order strategies would be more powerful in exploring label correlations (Read et al., 2009; Zhang & Zhou, 2014).

### 3.3 THREAT MODEL

In existing defense approaches such as adversarial training and defensive distillation, a threat model only needs to define the capabilities of attackers because those approaches focus on improving the generalization and thus robustness of the machine learning models that perform the original classification task. Typically, two types of attacks are defined: (1) *white-box attacks* that have the full knowledge about the original machine learning model including its architecture, parameter values, and training data, and thus can replicate the exact model to generate adversarial examples, and (2) *black-box attacks* that only have the limited knowledge about the original machine learning model such as the labels for a given set of input examples, and thus often need to create another "substitute model" to generate adversarial examples (Papernot et al., 2016a; Goodfellow et al., 2018).

In our defensive distinction approach, a thread model needs to further consider the capabilities of defenders because the effectiveness of our approach is related to the defenders' knowledge about the machine learning model (including its architecture, parameter values, and training data), adversarial example generation methods, parameters (e.g., maximum distortion) of the generation methods, and source examples used by attackers. We refer to these factors as AdvGen-Model, AdvGen-Methods, AdvGen-Parameters, and Adv-Examples. These considerations together with the capabilities of attackers appear to complicate the overall threat model; however, starting directly from the perspective of defenders, we can still clearly define some representative scenarios and cases as shown in Table 1.

Table 1: Representative scenarios and cases in the threat model from the perspective of a defender regarding if some information about the attacker is known.

| | AdvGen-Model Known | | AdvGen-Model Unknown | |
| --- | --- | --- | --- | --- |
| | AdvGen-Parameters Known | AdvGen-Parameters Unknown | AdvGen-Parameters Known | AdvGen-Parameters Unknown |
| Adv-Examples Known | Scenario 1, Case 1 **(S1C1)** | Scenario 1, Case 3 **(S1C3)** | Scenario 2, Case 1 **(S2C1)** | Scenario 2, Case 3 **(S2C3)** |
| Adv-Examples Unknown | Scenario 1, Case 2 **(S1C2)** | Scenario 1, Case 4 **(S1C4)** | Scenario 2, Case 2 **(S2C2)** | Scenario 2, Case 4 **(S2C4)** |

Briefly, Scenarios 1 and 2 represent that an attacker's AdvGen-Model is known or unknown to a defender, respectively; Cases 1 to 4 represent the four combinations of the defender's knowledge on AdvGen-Parameters and Adv-Examples, respectively. In total, eight scenario-case combinations (from S1C1 to S2C4) or simply cases are considered in this paper. Note that we do not include AdvGen-Methods in the table because it is reasonable to assume that popular adversarial example generation methods are known to both defenders and attackers (similar to the basic assumption in cryptographic systems that algorithms should not be considered as secrets and could be known by both defenders and attackers); however, we do consider the AdvGen-Methods factor in some of our experiments in the next section by leaving out certain generation method in model training.

---

[2]It can also be merged with the DD-Primary model to create a more complex model as described in Section 3.1 corresponding to the situation when $n > 2$.

## 4 EXPERIMENTS

We now evaluate the effectiveness of our defensive distinction approach by performing three sets of experiments with two for DDP-Model and one for DDS-Model.

### 4.1 DESIGN AND SETUP OF EXPERIMENTS

We design our experiments by using the MNIST dataset of handwritten digits and two different convolutional neural networks (CNNs). The first CNN is a slight variation of the original LeNet-5 (Lecun et al., 1998) by adding rectified linear units to the convolutional layers and using max pooling in the sub-sampling layers. We refer to this network as LeNet-5. The second CNN network has a convolutional layer with 64 filters followed by two convolutional layers with 128 filters, one 2 by 2 max pooling layer, one fully connected layer with 64 rectified linear units, and one softmax layer. We refer to this network as Basic-CNN. When we train these two CNNs for different classification tasks in our experiments, we use learning rate $\eta$=0.001, number of epochs=6, and batch size=128. For the original 10-digit classification task, LeNet-5 and Basic-CNN achieve 99.05% and 98.99% accuracy, respectively, on the 10,000 images from the MNIST test set.

In the generation of adversarial examples, one LeNet-5 network trained with the 60,000 images from the MNIST training set is considered as the attacker's model (AdvGen-Model) and six generation methods including FGSM, JSMA, BIM, L-BFGS, MIM, and PGD (Section 2) are used by leveraging the v2.1.0 of the CleverHans library (Papernot et al., 2018). Based on each source example (representing one digit) chosen from the MNIST test set, nine target adversarial example generation attempts (for the rest nine digit classes) are made. In other words, we consider targeted attacks, which are more damaging than untargeted attacks that simply misclassify source examples.

We generate five adversarial example datasets using the six methods based on different method parameters and source examples. As shown in Table 2, datasets adv_tr and adv_C1 are generated by using the same source examples with the index range 0-299 for each digit class such as '0', i.e., the first 300 examples from each digit class in the MNIST test set are chosen as source examples. These two datasets also share the same method parameters, but they are different due to the difference in the initial randomization of their generation. Dataset adv_C2 has the same method parameters but different source examples compared with adv_tr. Dataset adv_C3 has the same source examples but different method parameters compared with adv_tr. Dataset adv_C4 has different source examples and different parameters compared with adv_tr. The total number of adversarial examples in each dataset is approximately 300×9×6=16,200.

Table 2: Five adversarial example datasets. Source example indices are for each digit class. Method parameters notations: $\epsilon$ is input variation, $\theta$ is pixel step size, $\gamma$ is maximum distortion, max_iter is the maximum number of iterations, and step is linear search step size.

| Notation | Source Examples | FGSM $\epsilon$ | JSMA $\theta, \gamma$ | BIM $\epsilon$ | L-BFGS max_iter, step | MIM $\epsilon$ | PGD $\epsilon$ |
|---|---|---|---|---|---|---|---|
| adv_tr / adv_C1 | 0-299 | 0.3 | 1, 0.1 | 0.3 | 1000, 5 | 0.3 | 0.3 |
| adv_C2 | 300-599 | 0.3 | 1, 0.1 | 0.3 | 1000, 5 | 0.3 | 0.3 |
| adv_C3 | 0-299 | 0.25 | 0.5, 0.1 | 0.25 | 500, 5 | 0.25 | 0.25 |
| adv_C4 | 300-599 | 0.2 | -0.5, 0.1 | 0.2 | 1000, 10 | 0.2 | 0.2 |

Adversarial examples in adv_tr will be used for training our DDP-Model and DDS-Model defensive distinction models based on LeNet-5 and Basic-CNN. Models trained based on LeNet-5 represent the Scenario 1 in Table 1 because the AdvGen-Model also uses LeNet-5, while models trained based on Basic-CNN represent the Scenario 2 in Table 1. Adversarial examples in adv_C1 to adv_C4 will be used for testing our defensive distinction models, corresponding to the Cases C1 to C4 in Table 1. So now, our design and setup of experiments allow us to evaluate our defensive distinction approach for all the eight scenario-case combinations from S1C1 to S2C4 (Table 1).

### 4.2 DDP-MODEL WITH ADVERSARIAL EXAMPLES GENERATED BY A SINGLE METHOD

The simplest DDP-Model is a binary classifier for distinguishing natural examples from adversarial examples generated by a single attack method. For each method, a classifier is trained on a mixture

of 1,000 natural examples from the MNIST training set and 1,000 adversarial examples from the dataset adv_tr in Table 2 corresponding to the selected method. The classifier is then evaluated on a mixture of 1,000 natural examples from the MNIST test set and 1,000 adversarial examples from one test dataset adv_C1 to adv_C4 in Table 2 corresponding to the selected method. Note that in this paper, examples are all randomly selected from the corresponding sets; each experiment including the training and testing is independently performed for 10 runs to present the average results.

Model accuracy results for the four test cases with known AdvGen-Parameters (S1C1, S1C2, S2C1, S2C2) are given in Figure 1a. Classification accuracy is very high for all the six methods, suggesting that regardless of the defender's knowledge on AdvGen-Model (S1 vs. S2) and AdvGen-Examples (C1 vs. C2), a simple DDP-Model can be highly effective as long as AdvGen-Parameters are known. Across the four test cases, the models for distinguishing JSMA examples from natural examples struggled the most but still achieved high average accuracy at $92.35\%$, while the models for other methods such as MIM, PGD, FGSM, and BIM classified examples perfectly. One reason for the imperfect results on the JSMA examples could be that JSMA only selects a small number such as a pair of input features (i.e., pixels) for perturbation in each iteration (Papernot et al., 2016a).

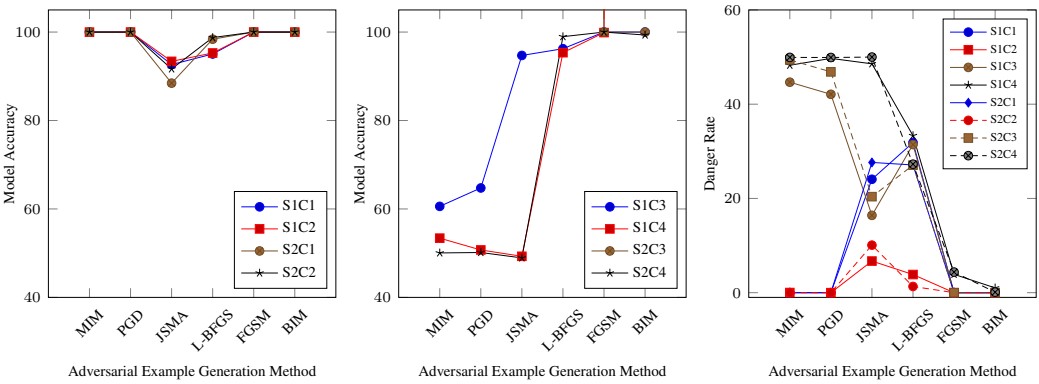

(a) Accuracy for S1C1, S1C2, S2C1, S2C2.  (b) Accuracy for S1C3, S1C4, S2C3, S2C4.  (c) Danger rate of successful adv. examples.

Figure 1: DDP-Model with adversarial examples generated by a single method: model accuracy is in (a) and (b), and danger rate of successful adversarial examples is in (c).

Model accuracy results for the remaining four test cases with unknown AdvGen-Parameters (S1C3, S1C4, S2C3, S2C4) are given in Figure 1b. Across the four test cases, classification accuracy is still very high ($> 95\%$) for L-BFGS, FGSM, and BIM, suggesting that a simple DDP-Model can still be highly effective for these three methods even when some AdvGen-Parameters are unknown to a defender. Models for the other three methods did not perform well. Notably, the models only achieved near $50\%$ accuracy (i.e., like random guessing) for MIM, PGD, and JSMA examples under the test cases S1C4 and S2C4, and only performed slightly better than random guessing for MIM and PGD examples under the test cases S1C3 and S2C3; these results suggest that knowing the parameters of some adversarial example generation methods is important to defenders.

Overall, the results in Figures 1a and 1b are highly informative in practice for defenders. Most importantly, it is always beneficial for defenders to train a DDP-Model by using adversarial examples generated based on a variety of parameters. Meanwhile, the exact model architecture and the exact natural examples used by attackers are not influential in the accuracy of the defenders' models.

So far, we analyzed the accuracy of DDP-Model without considering whether an adversarial example successfully fooled the Original-Model for the 10-digit classification task to misclassify it as a targeted digit class. Back to our multi-label classification formulation of the problem in Section 3.1, correlating the labels of these two models on an adversarial example will be interesting and will indeed further demonstrate the value of our approach. Especially, we define a *Danger Rate* metric, which is the percentage of the successful adversarial examples (i.e., those that achieved their targeted attacks) that are not classified by our DDP-Model as adversarial and thus will continue to incur danger. Without using DDP-Model, the danger rate of successful adversarial examples is considered as 100%. Figure 1c demonstrates that our DDP-Model can significantly reduce the danger rate of successful adversarial examples by over 50% for all the eight test cases and all the six methods.

### 4.3 DDP-MODEL WITH ADVERSARIAL EXAMPLES GENERATED BY MULTIPLE METHODS

A slightly more advanced DDP-model is a binary classifier for distinguishing natural examples from a collection of adversarial examples generated by multiple attack methods. Since it is unreasonable to assume that a defender will always know in advance all the methods used by an attacker, we introduce the concept of an excluded or "left-out" method in the experiments to consider the AdvGen-Methods factor. Basically, training is on adversarial examples selected from five attack methods with one method left out, while testing is on adversarial examples selected from all the six methods. We rotate the left-out attack method to explore the differences between methods, and compare the accuracy of the left-out classifiers with that of baseline classifiers which do not leave out a method.

A left-out classifier is trained on a mixture of $5,000$ natural examples from the MNIST training set and $1,000$ adversarial examples by each of the five non-excluded attack methods from the dataset adv_tr in Table 2. A baseline classifier is trained on a mixture of $6,000$ natural examples from the MNIST training set and $1,000$ adversarial examples by each of the six attack methods from the dataset adv_tr. They are all evaluated on $6,000$ natural examples from the MNIST test set and $1,000$ adversarial examples by each of the six attack methods from one test dataset adv_C1 to adv_C4.

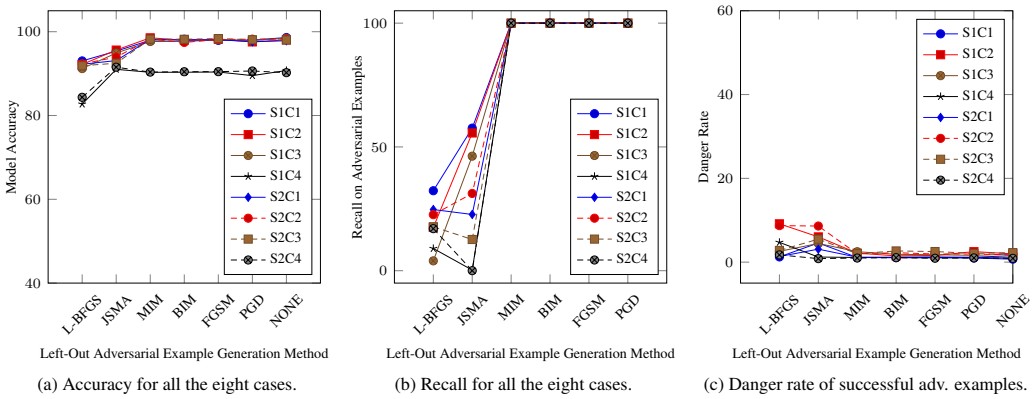

(a) Accuracy for all the eight cases.  (b) Recall for all the eight cases.  (c) Danger rate of successful adv. examples.

Figure 2: DDP-Model with adversarial examples generated by multiple methods: model accuracy is in (a), recall on the left-out adversarial examples is in (b), and danger rate is in (c).

Accuracy results for the left-out and baseline classifiers are shown in Figure 2a, where the x-axis identifies the left-out method (with *NONE* for baseline) in the training of a classifier, and the y-axis displays the overall accuracy of a classifier on a collection of $6,000$ adversarial examples and $6,000$ natural examples. In several of the test cases (S1C1, S1C2, S1C3, S2C1, S2C2, S2C3), classification accuracy is above $90\%$ (and often even well above $95\%$) across all methods. In the remaining test cases S1C4 and S2C4, classification accuracy is lower but still above $80\%$ across all methods.

Figure 2b further details the recall (or sensitivity) of a given classifier on a set of $1,000$ adversarial examples generated by the left-out method. We can see that clear disparities exist between two groups of methods: L-BFGS and JSMA examples well evade the classification when the corresponding method was not explicitly considered in the training, while the examples of the other four methods are accurately classified as adversarial even when the corresponding method was not considered in the training. These disparities can be explained to a certain extent by the fact that BIM, MIM, and PGD are all fine-grained refinements over FGSM as reviewed in Section 2. Figure 2c further demonstrates that our models can significantly reduce the danger rate of successful adversarial examples by over 90% (and often over 95%) for all the eight test cases and all the six methods.

Overall, Figure 2 clearly demonstrates that a DDP-Model trained on adversarial examples generated by multiple methods can be highly effective. While it is always beneficial for defenders to train a DDP-Model using adversarial examples generated by as many as possible methods, our models do have the strong capability to correctly classify the adversarial examples of an unknown method whose siblings are known as shown in the results for the four methods FGSM, BIM, MIM, and PGD that belong to the same family. In addition, similar to what we have observed in Section 4.2, knowing the parameters of some adversarial example generation methods is still important to defenders.

### 4.4 DDS-MODEL WITH ADVERSARIAL EXAMPLES GENERATED BY MULTIPLE METHODS

DDS-Model aims to distinguish known adversarial examples as generated by different specific methods. It can inform a defender about the specific techniques used by an attacker, and further help the defender exploit the weaknesses of the attack techniques. We trained a classifier using 1,000 adversarial examples generated by each of the six methods for a total of 6,000 examples from the dataset adv_tr in Table 2. We then evaluated the classifier using 1,000 adversarial examples generated by each of the six methods for a total of 6,000 examples from one test dataset adv_C1 to adv_C4.

Model accuracy results for the four test cases with known AdvGen-Parameters are shown in Figure 3a. With the exception of the MIM and PGD methods, adversarial examples from other methods are all accurately ($> 90\%$) classified across the four test cases. Model accuracy results for the remaining four test cases with unknown AdvGen-Parameters are provided in Figure 3b. Across these four test cases, PGD and MIM completely evaded our classifiers and FGSM was classified with low accuracy, but the remaining three methods (except for JSMA in case S1C4) were classified accurately. Figure 3c shows the full confusion matrix summed from the 10 runs of each experiment for S1C1, while the confusion matrices for all the eight cases are provided in Appendix B. The corresponding confusion matrices for the later four cases can help explain why PGD and MIM are the most difficult methods for correct classification – theirs examples are largely classified as BIM examples; it is presumably because while PGD starts with random perturbation (Madry et al., 2018) and MIM integrates momentum to escape local maxima (Dong et al., 2018) thus both resulting in diverse examples, they still resemble their close sibling BIM (Kurakin et al., 2017b) very much.

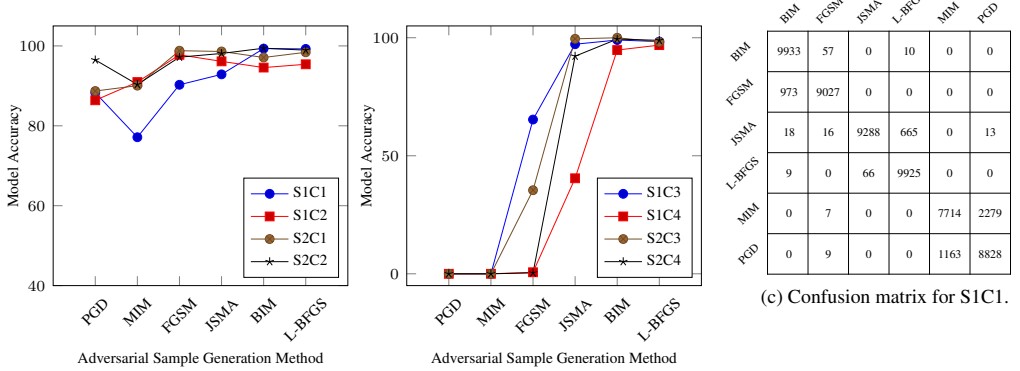

(a) Accuracy for S1C1, S1C2, S2C1, S2C2. (b) Accuracy for S1C3, S1C4, S2C3, S2C4.

(c) Confusion matrix for S1C1.

Figure 3: DDS-Model with adversarial examples generated by multiple methods: model accuracy is in (a) and (b), the full confusion matrix for the case S1C1 is in (c).

Overall, these results demonstrate the strong distinguishability of competing adversarial example generation methods. Similar to what is observed for a DDP-Model in Sections 4.2 and 4.3, it is also beneficial for defenders to train a DDS-Model by using adversarial examples generated based on a variety of parameters; meanwhile, the exact model architecture and the exact natural examples used by attackers are not influential in the accuracy of the defenders' models.

## 5 CONCLUSION

We proposed two important research questions that concern the distinguishability of adversarial examples, and formulated a classification problem in adversarial environments as a multi-label classification problem. We proposed a defensive distinction protection approach to answer the two questions and address the problem. We designed and performed experiments using the MNIST dataset and eight representative cases. Our experimental results demonstrate the strong distinguishability of adversarial examples, and the practical as well as research value of our approach. Our work also suggests many possibilities for the future work such as adopting high-order multi-label learning strategies to further explore the intrinsic correlations of labels as discussed in Section 3.2, investigating the distinguishability of adversarial examples for large tasks such as on ImageNet, and investigating the appropriate ways for integrating defensive distinction with other defense approaches.

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

## A  APPENDIX: A POTENTIAL EXTENSION TO THE PROBLEM FORMULATION

More labels could be added to include more concepts or semantic meanings in our multi-label classification formulation of the problem. For example, $y'_i$ can be extended to a triplet $(a_i, b_i, c_i) \in \mathbb{Y}'$ where $\mathbb{Y}' = \mathbb{Y} \times \mathbb{Z} \times \mathbb{Y}$ is a 3-ary Cartesian product, and $c_i \in \mathbb{Y}$ can indicate the source example class from which the input example $x_i$ was created. In the training set, $c_i$ can simply be $a_i$ for a natural example, and is assumed to be known for an adversarial example. This more complex version of formulation has its values on further correlating to the labels of source examples, but we do not explore it in the paper.

## B  APPENDIX: CONFUSION MATRICES FOR DDS-MODEL

|        | BIM  | FGSM | JSMA | L-BFGS | MIM  | PGD  |
|--------|------|------|------|--------|------|------|
| BIM    | 9933 | 57   | 0    | 10     | 0    | 0    |
| FGSM   | 973  | 9027 | 0    | 0      | 0    | 0    |
| JSMA   | 18   | 16   | 9288 | 665    | 0    | 13   |
| L-BFGS | 9    | 0    | 66   | 9925   | 0    | 0    |
| MIM    | 0    | 7    | 0    | 0      | 7714 | 2279 |
| PGD    | 0    | 9    | 0    | 0      | 1163 | 8828 |

(a) S1C1

|        | BIM  | FGSM | JSMA | L-BFGS | MIM  | PGD  |
|--------|------|------|------|--------|------|------|
| BIM    | 9458 | 540  | 0    | 2      | 0    | 0    |
| FGSM   | 223  | 9777 | 0    | 0      | 0    | 0    |
| JSMA   | 13   | 0    | 9613 | 369    | 0    | 5    |
| L-BFGS | 13   | 0    | 447  | 9540   | 0    | 0    |
| MIM    | 0    | 1    | 0    | 0      | 9095 | 904  |
| PGD    | 0    | 1    | 0    | 0      | 1360 | 8639 |

(b) S1C2

|        | BIM  | FGSM | JSMA | L-BFGS | MIM | PGD |
|--------|------|------|------|--------|-----|-----|
| BIM    | 9903 | 57   | 1    | 39     | 0   | 0   |
| FGSM   | 3466 | 6534 | 0    | 0      | 0   | 0   |
| JSMA   | 1    | 5    | 9725 | 248    | 0   | 21  |
| L-BFGS | 36   | 0    | 127  | 9837   | 0   | 0   |
| MIM    | 8765 | 1234 | 1    | 0      | 0   | 0   |
| PGD    | 9853 | 136  | 0    | 11     | 0   | 0   |

(c) S1C3

|        | BIM  | FGSM | JSMA | L-BFGS | MIM | PGD |
|--------|------|------|------|--------|-----|-----|
| BIM    | 9478 | 0    | 1    | 521    | 0   | 0   |
| FGSM   | 9926 | 65   | 0    | 9      | 0   | 0   |
| JSMA   | 0    | 0    | 4046 | 5954   | 0   | 0   |
| L-BFGS | 28   | 0    | 285  | 9687   | 0   | 0   |
| MIM    | 9917 | 9    | 1    | 73     | 0   | 0   |
| PGD    | 9628 | 2    | 2    | 368    | 0   | 0   |

(d) S1C4

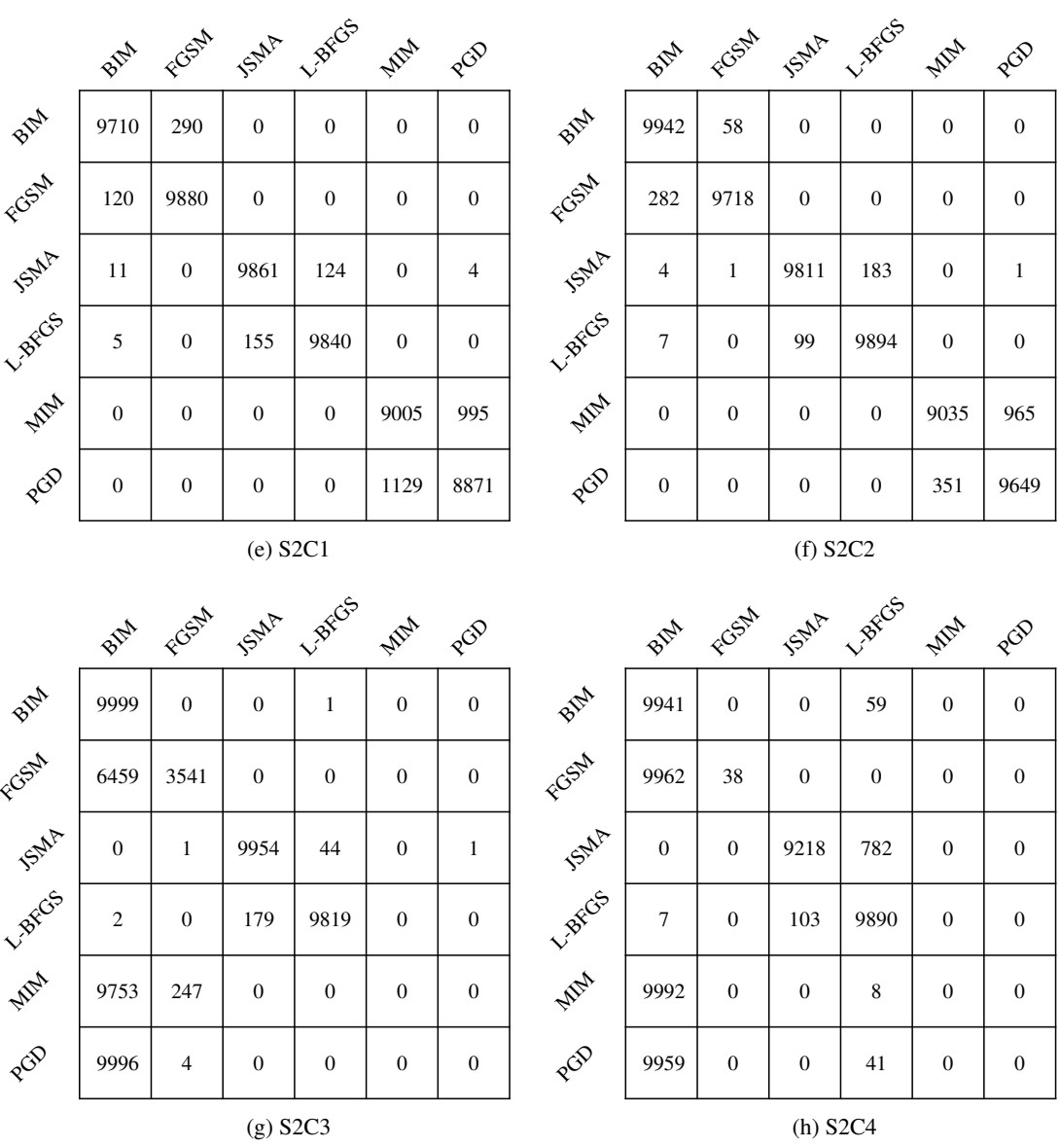

Figure 4: Confusion matrices for DDS-Model.

