# OpenReview forum: "Distinguishability of Adversarial Examples"
_ICLR.cc/2019/Conference_

### Official Review · AnonReviewer3 · 2018-10-31
**Interesting experiments but with major questions on defensive distinction.**

**Rating:** 4
**Confidence:** 4

**Review:**

In this paper, the authors proposed 'defensive distinction' to address questions: Are adversarial examples distinguishable from natural examples? Are adversarial examples generated by different methods distinguishable from each other?

I have some major concerns about this submission.

1) The presentation of this work should further be improved. It contains many vague sentences. For example, "Unfortunately, even state-of-the-art defense approaches such as adversarial training and defensive distillation still suffer from major limitations and can be circumvented." I really hope I can see some justifications based on authors' approach for this argument. Also, the definition of 'AdvGen-Model' is not clear. Do you mean Adversarial attack generator knows the network model (i.e., white-box attack)? It is also not clear that how representative scenarios and cases in Table 1 affect the implementation of the proposed experiments (implementation details rather than results).

2) The technical contribution of this paper is weak, and the experiments are not enough to support its main claim. MNIST is a simple dataset, please try larger and more complex datasets. The contribution of the current version is limited.

---

### Official Review · AnonReviewer2 · 2018-11-06
**Not enough depth**

**Rating:** 4
**Confidence:** 5

**Review:**

Defensive Distinction (DD) is an interesting model for detecting adversarial examples. However, it leaves some key aspects of defense and distinction out. Firstly, one can argue that if you know the adversaries of your model you can simply regularize the model for them. Even if regularization doesn't work fully, the DD model still suffers since it can have its own adversarial examples. From distinction perspective, it would be hard to believe that every single adversarial example will be detected, at least not without some solid theoretical background. It seems that  and natural examples are being thrown at the DD model without an elegant approach.

I have the following concerns about the visualization and understanding of what DD does, which I believe should have been the focus of this paper. It was not immediately clear, what the message of the paper is or the claimed message was too weak: detecting adversarial examples using a classifier. It was not immediately clear why this is a good idea (since an adversarial example can be an adversary of both original network and DD) or what the DD learns.

Furthermore, from experimental perspective, it is not sufficient to just perform experiments on one dataset, specially if the claim is big. You should consider running your model on multiple datasets and reporting what each DD learned. Furthermore, you should establish better comparison and back your claims with proper references. Some claims were too strong to believe without reference.

I do look forward to seeing more about the visualization and intriguing properties which may arise from continuation of your studies. In the current state, I vote to reject until a more clear demonstration of your work comes out.

---

### Official Review · AnonReviewer4 · 2018-11-12
**Good research direction, but needs more datasets**

**Rating:** 4
**Confidence:** 4

**Review:**

Summary: The authors propose two research questions: (1) Are adversarial examples distinguishable from natural examples? And (2) are adversarial examples generated by different methods distinguishable from each other? They find positive answers to both questions according to their experiments, and propose a method for detecting adversarial examples.

The authors take the viewpoint of varying how much the defender knows about its attackers. How they define whether a defender “knows” an attackers’ model, source examples, or adversarial generation parameters, is through keeping characteristics of various test sets the same with the training set. For example, to test the effectiveness of when the defender “knows” the adversarial generation parameters, they will have a test set where the adversarial generation parameters are the same with the training set, but will possibly vary other characteristics. They do all their experiments on MNIST.

In the first experiment (Section 4.2), the authors find that a deep neural network binary classifier for detecting adversarially-tainted images does well when the adversarial generation parameters are known, and not as well when unknown. Thus, the author’s conclude “it is always beneficial for defenders to train a DDP-Model by using adversarial examples generated based on a variety of parameters. Meanwhile, the exact model architecture, and the exact natural examples used by attackers are not influential in the accuracy of the defenders’ models.”

In the second experiment (Section 4.3), the authors test whether a neural network is able to classify an image as adversarial if images from a particular adversarial generation method is left-out of the training samples, but all others are included. They conclude that the network has the hardest time when samples from L-BFGS and JSMA are left-out of the training sample.

In the last experiment (Section 4.4), the authors test whether a deep neural network can classify adversarially-generated images according the the generation method. The answer is affirmative, and they conclude, “Similar to what is observed for a DDP-Model in Section 4.2 and 4.3, it is also beneficial for defenders to train a DDS-Model by using adversarial examples generated based on a variety of parameters; meanwhile, the exact model architecture and the exact natural examples used by attackers are not influential in the accuracy of the defender’s models.”


Stengths: The authors’ research questions are interesting and worthy of more investigation, namely whether we can detect adversarial examples. They also have nice experiments and make nice heuristic conclusions.


Weaknesses: The main complaint I have is that the authors only use the MNIST dataset. And we know that the MNIST dataset is special, so I would have liked to see the same tests on different datasets, and possibly different model architectures. I think this will be a much better contribution to the field with these additions.


Other comments:
The paper is clearly written and their experimental methodology seems original, and examining whether adversarial examples can be distinguished from untainted examples is important. But only using MNIST currently severely lowers the significance of this work. I think with more datasets and perhaps different model architectures, this can become a nice contribution to the field.

Perhaps a minor point, but their terminology of “natural” might not be the best, as MNIST is not usually considered as a “natural image,” although I am aware that what the author say is “natural” means “original,” or “untainted”. I would maybe suggest the authors change this terminology.

---

### Public Comment · (anonymous) · 2018-10-30
**Comparison to past work**

Much prior work has tried to distinguish adversarial examples from clean images. See, for example, Metzen et al. 2017, Grosse et al. 2017, Li et al. 2017. None of these approaches turned out to be effective (Carlini & Wagner 2017). How does your approach compare with those, which all propose training on adversarial examples in order to detect them.

In particular, do you attempt to actively evade your detection scheme?


Jan Hendrik Metzen, Tim Genewein, Volker Fischer, and Bastian Bischoff. On Detecting Adversarial Perturbations. In International Conference on Learning Representations (2017).
Kathrin Grosse, Praveen Manoharan, Nicolas Papernot, Michael Backes, and Patrick McDaniel. On the (Statistical) Detection of Adversarial Examples. arXiv preprint arXiv:1702.06280 (2017).
Xin Li and Fuxin Li. 2016. Adversarial Examples Detection in Deep Networks with Convolutional Filter Statistics. arXiv preprint arXiv:1612.07767 (2016).
Nicholas Carlini and David Wagner. Adversarial Examples Are Not Easily Detected: Bypassing Ten Detection Methods. arXiv preprint arXiv:1705.07263 (2017).

---

> ### Author Response · Authors · 2018-10-31
> **RE: Comparison to past work**
>
> Thank you very much for your questions and for pointing out the related work.  Our original focus was intensively on the comparison of different attack methods.  We switched our focus to the defense side and completed this work without being aware of these related papers (very sorry about this).  We will cite them and compare them with ours in the revision.
>
> In short, our approach is under the formulation of multi-label classification, which is different from those in the related work.  We discussed the important advantages of this approach such as exploring the intrinsic correlations between multiple labels in Section 3.1.  In addition, we explored a DDS-Model for distinguishing known adversarial examples as generated by different specific methods.
>
> We just read the C&W paper “Adversarial Examples Are Not Easily Detected …”, and find it very interesting and helpful.  We completely agree that adversarial examples are not easily detected.  Coincidentally and related to our original focus on attack comparison, we defined eight representative scenario-case combinations in Table 1 of our paper and constructed five datasets (Table 2) to extensively identify the situations in which adversarial examples are (vs. are not) easily detected. Our approach is more intensively tested (with both weak and strong attack methods) than some of the approaches evaluated by Carlini & Wagner. At the end of each experiment section (4.2, 4.3, 4.4), we highlighted the following key observations that are helpful to the defenders:
>
> (1) “Overall, the results in Figures 1a and 1b are highly informative in practice for defenders. Most importantly, it is always beneficial for defenders to train a DDP-Model by using adversarial examples generated based on a variety of parameters. Meanwhile, the exact model architecture and the exact natural examples used by attackers are not influential in the accuracy of the defenders’ models.”
>
> (2) “Overall, Figure 2 clearly demonstrates that a DDP-Model trained on adversarial examples generated by multiple methods can be highly effective. While it is always beneficial for defenders to train a DDP-Model using adversarial examples generated by as many as possible methods, our models do have the strong capability to correctly classify the adversarial examples of an unknown method whose siblings are known as shown in the results for the four methods FGSM, BIM, MIM, and PGD that belong to the same family. In addition, similar to what we have observed in Section 4.2, knowing the parameters of some adversarial example generation methods is still important to defenders.”
>
> (3) “Overall, these results demonstrate the strong distinguishability of competing adversarial example generation methods. Similar to what is observed for a DDP-Model in Sections 4.2 and 4.3, it is also beneficial for defenders to train a DDS-Model by using adversarial examples generated based on a variety of parameters; meanwhile, the exact model architecture and the exact natural examples used by attackers are not influential in the accuracy of the defenders’ models.”
>
> Back to your specific question “do you attempt to actively evade your detection scheme?”.  Indeed, it is considered in our attack model and experiments as you might already perceive based on the observations we listed above.  We defined the threat model from the perspective of a defender as mentioned in Section 3.3.  Whether a defender knows the network model, parameters, source examples, and attack methods (i.e., the “left-out” experiments in Section 4.3) used by attackers is equivalent to a large extent to how an attacker may evade the detection.  The observations listed above are also important suggestions for defenders to mitigate the evasions.  In other words, based on our preliminary analysis, our eight scenario-case combinations in Table 1 and the “left-out” experiments in Section 4.3 include and indeed further extend the three types of attacks (“Zero-Knowledge”, “Perfect-Knowledge”, and “Limited-Knowledge”) considered in the C&W paper.
>
> It is always an arms race between attackers and defenders, no matter a defense solution takes the adversarial training approach or the adversarial example detection approach.  Even considering the attacks in the C&W paper, it is reasonable to assume that defenders may know the details of the attacks and will further improve the detection accuracy.  On the other hand, attackers can always leverage their knowledge of defense solutions to generate new adversarial examples to fool the systems. Thank you again for reading our paper and providing valuable comments.

---

### Author Response · Authors · 2018-11-17
**Proposed Changes to Paper**

We would like to thank the reviewers for their valuable suggestions and comments. We really appreciate all the feedback from the reviewers as well as readers. We believe this has benefited our revision.

Based on the reviews, one common concern is the dataset. Specifically, results predicated on the MNIST dataset alone are not convincing enough because the MNIST dataset is special. We plan to strengthen our results by applying our approach in two other domains (CIFAR10 and ImageNet-10) using different model architectures (ResNet and VGG16). Would these additional datasets and architectures be sufficient for the evaluation?

We also plan to address some of the more detailed critiques of our paper. We would greatly appreciate any other suggestions and comments concerning our proposed changes.

---

### Meta-Review · Area_Chair1 · 2018-12-18
**Interesting direction, but more work needed**

**Confidence:** 4
**Recommendation:** Reject

**Metareview:**

The paper investigates an interesting question and points at a promising research direction in relation to whether adversarial examples are distinguishable from natural examples.

A concern raised in the reviews is that the technical contribution of the paper is weak. A main concern with the paper is that the experiments have been conducted only on one simple data set. The authors proposed to add more experiments and improve other points, but a revision didn't follow.

The reviewers consistently rate the paper as ok, but not good enough.

I would encourage the authors to conduct the improvements proposed by the reviewers and the authors themselves.